# Assessing the Importance of Native Mycorrhizal Fungi to Improve Tree Establishment after Wildfires

**DOI:** 10.3390/jof9040421

**Published:** 2023-03-29

**Authors:** Cristian Atala, Sebastián A. Reyes, Marco A. Molina-Montenegro

**Affiliations:** 1Instituto de Biología, Facultad de Ciencias, Pontificia Universidad Católica de Valparaíso, Av. Universidad 330, Curauma, Valparaíso 2340000, Chile; 2Centre for Integrative Ecology (CIE), Instituto de Ciencias Biológicas, Universidad de Talca, Campus Lircay, Avda. Lircay s/n, Talca 3460000, Chile; 3Centro de Investigación en Estudios Avanzados del Maule (CIEAM), Universidad Católica del Maule, Talca 3460000, Chile

**Keywords:** arbuscular mycorrhizal fungi, Mediterranean-type ecosystems (MTEs), sclerophyllous forest, restoration, wildfires

## Abstract

The Chilean matorral is a heavily threatened Mediterranean-type ecosystem due to human-related activities such as anthropogenic fires. Mycorrhizal fungi may be the key microorganisms to help plants cope with environmental stress and improve the restoration of degraded ecosystems. However, the application of mycorrhizal fungi in the restoration of the Chilean matorral is limited because of insufficient local information. Consequently, we assessed the effect of mycorrhizal inoculation on the survival and photosynthesis at set intervals for two years after a fire event in four native woody plant species, namely: *Peumus boldus*, *Quillaja saponaria*, *Cryptocarya alba*, and *Kageneckia oblonga*, all dominant species of the matorral. Additionally, we assessed the enzymatic activity of three enzymes and macronutrient in the soil in mycorrhizal and non-mycorrhizal plants. The results showed that mycorrhizal inoculation increased survival in all studied species after a fire and increased photosynthesis in all, but not in *P. boldus*. Additionally, the soil associated with mycorrhizal plants had higher enzymatic activity and macronutrient levels in all species except in *Q. saponaria,* in which there was no significant mycorrhization effect. The results suggest that mycorrhizal fungi could increase the fitness of plants used in restoration initiatives after severe disturbances such as fires and, consequently, should be considered for restoration programs of native species in threatened Mediterranean ecosystems.

## 1. Introduction

Mycorrhizal fungi form a symbiosis with approximately 92% of known vascular plant species, according to recent estimations [1]. This interaction has several functions for both participants. The more widely studied function is the nutritional one, in which symbiotic fungi allow plants a more efficient nutrient (particularly N and P) and water uptake, absorbing them from the soil and delivering them to the plant in exchange for carbohydrates [2] and lipids [3,4,5]. Moreover, the increased nutrient and water uptake favored by mycorrhization can result in increased photosynthesis [6,7], and ultimately, increased growth and survival [7,8].

Consequently, this symbiosis can impact plant and fungal fitness, biogeochemical cycles, plant and fungal diversity, and soil aggregation [9], being pivotal in the structuring of the community [10,11]. Indeed, plants and their associated soil biota interact via multiple feedback mechanisms modulating the community assembly and function [12]. Plants interact directly with mutualists such as mycorrhizal fungi that affect establishment and maintenance in the community [13], but there are also indirect feedbacks exerted by these microorganisms via the modification of soil nutrient cycling [14]. Thus, mycorrhizal fungi must be considered in the restoration plans for disturbed ecosystems. Previous several studies on a variety of ecosystems have shown positive restoration outcomes related to the consideration of the symbiosis in terms of increments in plant growth and biomass, as well as plant species richness and survival [13,15].

Mediterranean-type ecosystems (MTEs) are severely threatened worldwide due to anthropogenic factors, thus becoming a main conservation concern [16]. The Chilean matorral (including the sclerophyllous forests) is one of those MTEs and occurs in Central Chile from approximately 30° to 36° S [17]. This ecosystem is currently experiencing very significant anthropogenic disturbances, such as land-use changes and intense wildfires [18,19,20]. Worryingly, this plant community is concentrated close to 2/3 of the total Chilean population [21], but it is under-represented in the national system of protected areas [22,23]. This is despite it having the highest vascular plant richness in the country, including many endemic species [24,25].

As mentioned above, this MTE has suffered very intense and large-scale fires in recent years, all of anthropogenic origin [26,27,28]. Fires in Mediterranean areas are not uncommon, and in some MTEs are naturally occurring [29] and can have severe negative consequences for biodiversity [30,31]. Fires can also, depending on the duration and intensity, impact soil microorganisms, rapidly changing microbial communities [32], and can affect other physicochemical aspects of the soil [33,34]. These changes can alter plant communities severely, transforming community structure in the long term [35]. Fires usually occur during summer, and in Chile, the post-fire natural regeneration of native plants is limited, usually with advantages to introduce plant species that are adapted to natural fires, particularly those from Australia and California [36,37]. Most native woody plants lack specializations for post-fire germination, but some species can re-sprout [28,36]. In the long term, actions should be taken to ensure the restoration of the Chilean MTE and avoid its further degradation after fires. Restoration initiatives in Chile have had very limited success and are usually costly and inefficient [24,38] with Chile seemingly unable to fulfill international agreements in this regard.

One possible solution for the restoration of MTEs after fires is the use of mycorrhizal fungi. As stated above, the association with these soil microorganisms can increase nutrient uptake, and confer plants increased stress tolerance [9,39]. After fires, soils tend to become exposed and susceptible to erosion [40]. High radiation, low water retention/penetration, and high-water stress can severely limit germination and re-growth of natural vegetation post-fire [41,42]. Thus, mycorrhizal fungi can be a key element for the restoration process by increasing plant performance in stressful conditions, improving water and nutrient uptake [43], and improving biological [44] and physicochemical characteristics of the soil [45,46]. Current knowledge of mycorrhizal fungi diversity and native plant-fungi associations in the Chilean Mediterranean zone, and sclerophyllous forest in particular, is very limited [47]. Thus, the effect of the mycorrhizal fungi, and principally mycorrhizal symbiosis, on the physiological performance of native plants is seldom assessed and, consequently, their application in restoration is very difficult to evaluate.

The aim of this study is to test the effect of mycorrhizal fungi from the sclerophyllous forest on the survival of the dominant woody-species in this MTE after fire. Additionally, we evaluated the foliar photosynthesis as well as the enzymatic activities and content of macronutrient in the soil in mycorrhizal and non-mycorrhizal plants. These mycorrhizal fungi could be essential for seedling establishment and overall plant performance. Thus, information regarding the efficacy of mycorrhizal fungi application to the seedlings of studied plant species could be essential for conservation and future restoration initiatives in the MTE of Central Chile, especially after fires.

## 2. Materials and Methods

### 2.1. Study Site and Target Species

The study site was located in the place called “Fundo los Perales” (33°06′ S, 71°37′ W) located south-west of the city of Valparaíso, Chile. This place is characterized by a Mediterranean climate with cold and rainy winters as well as dry and warm summers [17]. In recent years, the area has experienced frequent and intense fires, all of anthropogenic origin, with severely negative impacts to the local diversity [26,27,28,31]. The flora in the area is typically Mediterranean and highly endemic [25]. The most conspicuous woody species present in the community are *Peumus boldus* (Monimiaceae), *Quillaja saponaria* (Quillajaceae), *Cryptocarya alba* (Lauraceae), and *Kageneckia oblonga* (Rosaceae). Thus, all four of these woody plant species were examined in this study. In addition, those four plant genera form mycorrhizal associations with arbuscular mycorrhizal fungi (AMF) consequently form the arbuscular mycorrhiza (AM) symbiosis type [48].

### 2.2. Root Sampling and Fungal Inoculum 

Roots from ten randomly selected individuals of each tree species were collected. All trees were similar in height (approximately 2.5–3.0 m) and the distance between them was at least 10 m, to ensure the independence of the samples. These samples were used as biological material to obtain the inoculum containing mycorrhizal structures (spores, hyphae, and vesicles) to be utilized in the field experiments. Each fungal inoculum was prepared using a solution of 75 g of ground root containing mycorrhizal structures (indicated above) in 100 mL of distilled water. Roots from all individuals per species used to obtain the inoculum were mixed in a single flask to obtain one inoculum mix per each of the four target species.

### 2.3. Assessment of the Survival and Photosynthesis in the Inoculated and Non-Inoculated Trees

The effect of experimental inoculation with a mycorrhizal inoculum mix (see above) on survival and net photosynthesis (A) was assessed at 0, 3, 6, 12, and 24 months after transplants. A total of 50 seedlings of each of the four tree species were selected for these measurements. Half of them were randomly assigned to the “with mycorrhiza” treatment (M+), and the other half was assigned to the “without mycorrhiza” treatment (M−). Individuals of each of these tree species were obtained from seeds collected at the study site. Seeds were germinated and maintained in a greenhouse under semi-controlled conditions (air temperature = 25 ± 3 °C and solar radiation = 1056 ± 96 μmol m^−2^s^−1^). At the start of the experiment, the seeds of all individuals were washed and hydrated with a mixture of NaCl (2%) and a systemic antifungal (Benlate) to avoid the presence of microorganisms. Subsequently, half of the seedlings were irrigated with 100 mL/day of the fungal inoculum solution (see above). Each seedling received its own inoculum mix by manual irrigation. Individuals allocated to the treatment without mycorrhizae, only received irrigation (tap-water) without inoculum. The inoculation procedure was repeated 3 times in a 21-day period to ensure symbiosis. A later validation of the effectiveness of the inoculation was conducted using staining methods and a light microscope in a subsample (5 roots from 5 individuals per treatment) previously to be allocated into the field. It was considered a successful inoculation treatment when more than 75% and less than 5% of the individuals showed some type of mycorrhizal structure. Thus, the individuals used in our studies and assigned to the M+ and M− treatment showed more than 80% and less than 5% of AM structures, respectively (Appendix A). The main AM structures recorded after inoculation were mainly hyphae coils, vesicles, and/or arbuscules.

Three-month-old plants were transplanted to 50-L plastic pots (one plant per pot) containing a mixture of sand/native soil (1:1 *v*/*v*) as substrate and irrigated every two days with 250 mL of tap water. All pots were specially rearranged every week to avoid the effect of asymmetric solar incidence due to the shape of the glasshouse roof. Plants were maintained in these semi-controlled conditions until they were five months-old, and then were transported to the field to assess the effect of inoculation of mycorrhiza on photosynthesis and the survival on tree species as well as on the soil nutrient properties.

In the field, for each tree species, five groups of five individuals (one per pot) with mycorrhiza were established and another five groups were set up without them (total n = 50 seedlings per species). The pots were arranged in an area of 250 × 250 m. Individuals were placed no less than 0.5 m apart from each other, to respect the typical lighting conditions of each species. The experiment lasted for two years, and net photosynthesis as well as survival percentage were recorded at 0, 3, 6, 12, and 24 months after transplant to the field. To assess whether AM presence can improve some fitness-related traits, net photosynthesis was measured on visually healthy leaves from the upper third of the plant. Measurements were made on the same individual at midday each time with an infrared gas analyzer (IRGA, Infra-Red Gas Analyzer, CIRAS-2, PP-Systems Haverhill, Amesbury, MA, USA). During measurements, the leaf chamber was settled at 360 ppm, 25 °C, and 1000 μmol m^−2^s^−1^. We compared the survival percentage of seedlings at the same times as the photosynthesis measurements were done. Mortality was assessed visually, assuring the absence of green tissue in the individual plants. In the next survival assessment, all plants were inspected again to corroborate/discard the initial evaluation.

### 2.4. Enzymatic Activity and Nutritional Effect of the Mycorrhizal Inoculation on the Soil

We tested the effect of mycorrhizae on nutritional properties of the soil by measuring soil activity of three key enzymes (β-Glucosidase, urease, and dehydrogenase). The β-glucosidase is involved in glucose degradation and increased carbon availability [49]. Urease is a key enzyme related to the efficiency of nitrogen assimilation [50]. Lastly, dehydrogenase participates in maintaining cellular homeostasis, protein degradation, and the control of reactive oxygen species (ROS) [51].

At 0, 6, and 24 months after the transplant of trees, we sampled 50 g of rhizospheric soil from plants with and without mycorrhizal fungi to measure enzyme concentration. Enzymatic activity was measured following previously described methods [52]. β-Glucosidase activity was measured using 0.5 g of soil with added 0.5 mL of 4-Nitrophenil-B-D-glucopyranoside 50 mM (PNG) as the enzymatic substrate. Activity of this enzyme was thus measured as μg of P-Nitrophenol (PNP) per gram of PNG produced per hour (g PNG g-1 PNP h-1). For the urease activity, we used 1 g of soil in a 0.64% *v*/*v* solution of urea to obtain the amount of resulting NH4+. This was done measuring the solution absorbance at 525 nm using a spectrophotometer. Results for urease were expressed as µg N-NH4 g-1 h-1. Finally, dehydrogenase activity was measured also using 1 g of soil per pot. For each sample (from each individual pot), we added 0.2 mL of a 0.4% *v*/*v* solution of 2-(p-Iodophenyl)-3-(p-Nitrophenyl)-5-Phenyl tetrazolium chloride (INT) as substrate. Thus, dehydrogenase activity was registered as µg of reduced iodonitrotetrazolium formazan per hour (INTF g-1 h-1).

Complementarily, soil content of macro-nutrient (N, P, K) was quantified in 6 samples of 200 g of soil in both inoculated (M+) and not inoculated (M−) plants. Analyses were performed with the samples collected at 24 months after the transplant of the plants. Nutrient measurements were conducted in the Centro Tecnológico de Suelos y Cultivos at Universidad de Talca, Talca, Chile.

### 2.5. Statistical Analyses

To compare the effect of mycorrhiza on survival and net photosynthesis along time, repeated-measurements ANOVAs were conducted. The assumptions of normality and homogeneity of variances were tested using the Shapiro–Wilk and Levene tests, respectively. In addition, we carried out a pairwise *t*-test using the Bonferroni correction method to compare the effect of mycorrhizal inoculation on soil enzymatic activity (dehydrogenase, β-glucosidase and urease) as well as for soil macro-nutrient contents (N, P, K). Differences were considered significant at *p* < 0.05. The data were analyzed using the R environment [53]. Figures for survival, photosynthesis, enzymes, and soil nutrients were made using the *tidyverse* R package [54]. Lastly, to characterize differences in enzymatic and nutritional traits between inoculated (M+) and non-inoculated (M−) plants with AMF, for each species we calculated Cohen’s f statistics as a measure of standard size effect (SES) using the *Cohen’s_f* function from the *effectsize* R package [55], at 95% CI. Negative and positive SES values indicate an increase and decrease in the enzymatic or nutritional traits caused by AMF presence.

## 3. Results

Association with mycorrhizal fungi increased survival in *Cryptocarya alba*, *Kageneckia oblonga*, *Peumus boldus*, and *Quillaja saponaria* (Table 1, Figure 1). In addition, the interaction Time × Treatment was significant for *C. alba* and *P. boldus*, indicating that although the survival decreased in both treatments with time, this pattern was remarkably in those treatments without the inoculation with mycorrhizae (Table 1, Figure 1a,c). Although, *K. oblonga* exhibited a greater survival percentage with the presence of mycorrhizae, a similar trend was maintained over time (Figure 1b).

Additionally, mycorrhizal fungi also increased photosynthesis in all species of trees (Figure 2). On the other hand, the interaction between Time × Treatment was significant for *C. alba*, *P. boldus*, and *Q. saponaria* (Table 2), since photosynthesis was increased with time but with higher intensity in those individuals with a presence of mycorrhiza (Figure 2a,d). Nevertheless, the photosynthesis for *P. boldus* was higher in the treatment without AM fungi (M−) in some measurements (Figure 2c). Contrarily, in *K. oblonga*, there were no statistical differences between the interaction Time × Treatment (Table 2), although the photosynthesis was higher in the treatment with AM fungi (M+) in almost all measurements (Figure 2b).

At last, association with mycorrhizal fungi increased soil enzymatic activity of dehydrogenase, β-glucosidase, and urease, in *C. alba*, *K. oblonga,* and *P. boldus*, but not in *Q. saponaria* (Table 3, Appendix A) after 24 months. Furthermore, we found significant differences in the effect of mycorrhizal treatments, time, and their interaction (Time × Treatment), in dehydrogenase, β-glucosidase and urease, for *C. alba*, *K. oblonga* and *P. boldus* (Table 3, Appendix A). Similarly, inoculation with mycorrhizal fungi significantly increased soil macronutrients, in *C. alba*, *K. oblonga,* and *P. boldus*, but not in *Q. saponaria* (Table 4, Appendix A) after 24 months of transplantation of trees.

In addition, we detected that overall enzymatic activity as well as macronutrients were higher in the soil of mycorrhizal plants (M+) compared to non-inoculated plants (M−). This was the same with *C. alba*, *K. oblonga,* and *P. boldus* and was evident for all three tested enzymes (dehydrogenase, β-glucosidase, and urease) and macronutrients (N, P, and K), showing a high effect size of mycorrhization on these soil traits (Table 4, Figure 3). Contrarily, we found no increase in the enzymatic activity or macronutrients in the soil of inoculated plants of *Q. saponaria*, where the effect size was nearly zero (Table 4, Figure 3).

## 4. Discussion

All tree species included in this study showed greater survival at the end of transplant experiments when inoculated with their own mycorrhizal fungi. Previous studies in other MTEs have shown a similar positive impact on plant survival when plants are inoculated with native mycorrhizal fungi [56,57], indicating that mycorrhizal fungi are a promising tool for the restoration of MTEs, and in sclerophyllous forests in particular. 

Additionally, the association with mycorrhizal fungi increased photosynthesis in three out of the four studied species, *Cryptocarya alba*, *Kageneckia oblonga*, and *Quillaja saponaria*, dominant tree species of the sclerophyllous forest of Central Chile [17,58]. This positive effect of mycorrhiza on photosynthesis has been found for other plant species [6,59] and could be related to increased stomatal conductance and water-use efficiency [6,60,61] and/or an increased photochemical efficiency [62], particularly in the dry conditions of the late spring–summer in MTEs, when fires are more likely to occur. In contrast, *Peumus boldus* decreased its photosynthesis when inoculated with mycorrhizal inoculum used in this experiment. This could be due to an incompatibility between this species and that specific fungal partner, or that the mycorrhizal association triggers signal pathways that result in reduced physiological performance. A negative effect of mycorrhizal fungi has been reported on highly colonized plants, when the mutualism can turn into parasitism with negative consequences for the plant depending on the fungal species and environmental conditions [63,64]. Alternatively, a negative effect of fungal endophytes on mycorrhizal fungi has been recorded, suggesting that competition may occur among them [65,66]. In fact, Liu et al. [67] showed that plants previously infected with fungal endophytes decrease their mycorrhizal colonization, negatively affecting the host plant-growth, depending on the nutritional status and fungal strain. In our case, *P. boldus* showed a frequency of mycorrhizal infection similar to any other tree species, but nearly double fungal endophytes (Appendix A). Thus, the high frequency of fungal endophyte could negatively affect the mycorrhizal inoculation, which in turn could help explain the relative low values of enzymatic activity, soil nutrients, and photosynthesis recorded in *P. boldus*.

On the other hand, enzymatic activity in the soil, evaluated in three key enzymes, increased in all mycorrhizal plants, except in *Q. saponaria* where only β-Glucosidase activity was influenced by mycorrhization. This result indicates that mycorrhization increases soil biological activity and likely available nutrients for the plant. This concurs with findings in previous studies that show an increased enzymatic activity and overall soil microbial activity in mycorrhizal plants [68]. These can produce a synergistic effect of mycorrhizal fungi and a beneficial bacteria present in the soil enhancing plant survival and performance, probably due to an increased nutrient availability and antibiotic effect against pathogenic microorganisms [44,69,70]. Indeed, the soil nutrients increased in three of four tree species assessed when inoculated with mycorrhizal. Those trees that showed an enhancement in the nutritional status of soil were the ones that showed an increased enzymatic activity. In fact, the only tree species (*Q. saponaria*) without any effect on most of the enzymes assessed was the same species without improvement in the nutritional status of soil. Thus, our results could suggest that mycorrhizal inoculation could modulate in a concerted way the enzymatic activity and nutritional status of soil for inoculated trees.

Mycorrhizal symbiosis has often been interpreted as a mutualistic association, but the effects on plants vary within a parasitic-mutualistic continuum [71,72]. The symbiosis might be established because it can help plants cope with stressful environments [73], and/or because the plant constitutes an important habitat for the mycobiont [74]. Here, we found that mycorrhizal fungi had a positive nutritional effect on the plant and that may account, at least in part, for the positive effect in survival and photosynthesis. However, the symbiosis, and its derived effects, may occur due to other drivers [75]. Positive effects of mycorrhizal fungi on plants have been widely reported [6,8,76,77]. Association with mycorrhizal fungi is related to increased water and nutrient uptake, physiological performance, growth, and tolerance to biotic and abiotic stress [6,8,73,76,77,78]. All of these factors can result in an increase in survival, especially, as mentioned above, in the dry summer of Central Chile.

Overall, those trees that exhibited improved nutritional and enzymatic status levels were the same ones that demonstrated an increase in survival. Thus, mycorrhizal symbiosis could be used as a tool for increasing restoration success after fires in the Chilean MTE. This is highly relevant since MTEs, including the MTE in Central Chile, are considered biodiversity hotspots [79]. Climate change, forestry with pyrogenic species, and a prolonged drought have increased dry biomass and the likelihood of fires in Central Chile [80,81,82], giving birth to one of the most severe fires registered in recent history in 2017 [83]. Initiatives to restore the degraded MTE in Central Chile after those intense and extensive fires have had limited success due to economic, ecological, and cultural reasons [38]. Here, we demonstrate that inoculation with mycorrhizal fungi could greatly enhance the likelihood of success at a relative low cost, solving at least two of the main issues limiting restoration in Chilean matorral. In fact, mycorrhizal plants have been shown to be a viable and successful strategy for the restoration of other MTEs [84,85,86]. Advances in identification and inoculum production could further promote the role of mycorrhizal fungi in restoration.

A key next step in moving forward would be to identify the mycorrhizal species associated with the plant species investigated here [87]. If these woody plants have a similar mycorrhizal species composition, it could suggest a common mycorrhizal network in the Chilean MTE [43]. Since the Chilean matorral is constantly threatened by human-related factors [16], particularly intense human-provoked fires, the great habitat loss and direct loss of biodiversity might be recovered by planting target species near existing plants that could serve as an inoculum source of beneficial fungi. This is promising since evidence indicates that local fungi are more beneficial than commercial inoculum for plant restoration [88]. Additionally, complementary strategies could be used to improve the restoration success in highly perturbated zones as those affected by wildfires. Recently, Gerrits et al. [15] performed a meta-analysis synthesizing data from field experiments and their respective reference ecosystems across four continents. In this study, it is evidenced that soil translocation—including the microorganisms—can be a more successful restoration method than only with the use of propagules or seedlings. Thus, inoculation with native mycorrhizal could be part of a more integrative method to carry out successfully different restoration initiatives.

Thus, the use of whole native soils containing some selected mycorrhizal fungi could be a viable strategy for improving the restoration success of woody species in the MTEs after fire events. In recent years, symbiotic fungi have shown a high potential as biotechnological tools to improve plant performance and survival, especially under stressful conditions [6,89,90]. In the dry conditions of the late spring and summer of MTEs, fires are likely to keep occurring with an ever-increasing frequency, especially considering climate change and current land-use changes [91]. Inoculation with mycorrhizal could be a key strategy to improve the survival of the seedlings of native species in much needed restoration and propagation initiatives—at least—in the Chilean MTE, especially considering international accords such as REDD+, Nationally Determined Contributions, Bonn Challenge, and the Initiative 20 × 20 (see [38]), as well as the global commitments on the conservation of biodiversity.

## Figures and Tables

**Figure 1 jof-09-00421-f001:**
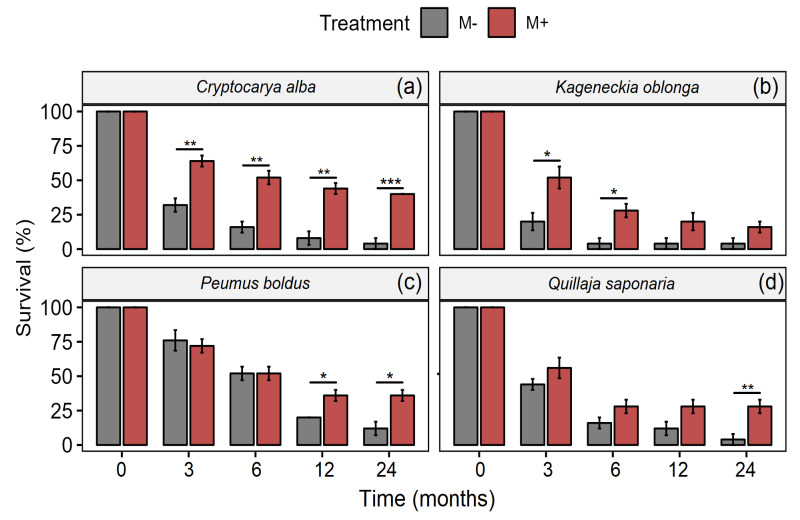
Survival in *C. alba* (**a**), *K. angustifolia* (**b**), *P. boldus* (**c**), and *Q. saponaria* (**d**), four dominant woody plant species from the Chilean Mediterranean ecosystem (MTE), after a fire with (red bars) and without (grey bars) inoculation with AMF. Values represent means (±1SE). Significant differences (pairwise *t*-test; Bonferroni correction) between treatments are represented with * *p* < 0.05; ** *p* < 0.01; *** *p <* 0.001.

**Figure 2 jof-09-00421-f002:**
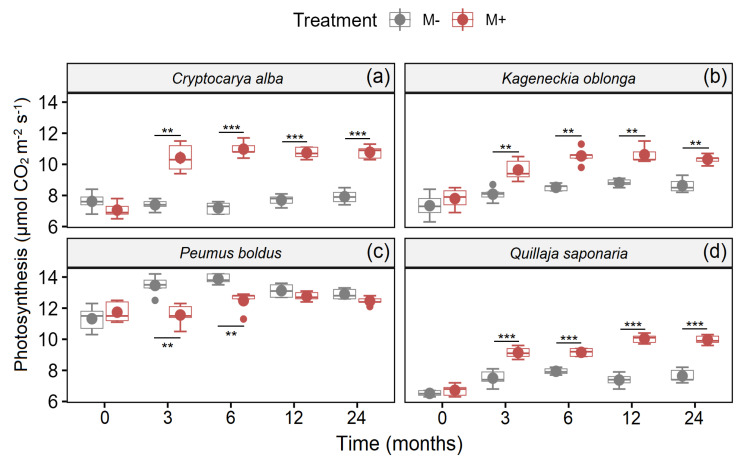
Photosynthesis in *C. alba* (**a**), *K. angustifolia* (**b**), *P. boldus* (**c**), and *Q. saponaria* (**d**), four dominant woody plant species from the Chilean Mediterranean ecosystem (MTE), after a fire with (red boxes) and without (grey boxes) inoculation with AMF. Values represent means (±1SE). Significant differences (pairwise *t*-test; Bonferroni correction) between treatments in all times measured are represented with ** *p* < 0.01; *** *p* < 0.001.

**Figure 3 jof-09-00421-f003:**
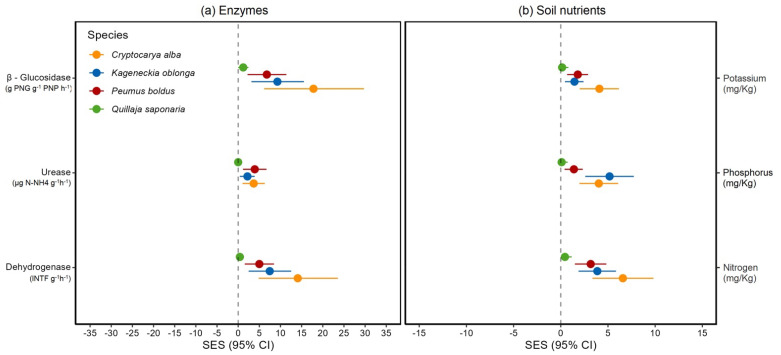
Effect size of (**a**) enzymatic activity and (**b**) soil nutritional content in four dominant woody plant species from the Chilean Mediterranean ecosystem (MTE), after a wildfire inoculated (M+) and not inoculated (M−) with AMF. SES with 95% CI measured after 24 months for urease, β–glucosidase, and dehydrogenase are shown. A mean effect size is significantly different from zero when CIs do not overlap zero. Negative (or positive) effect sizes indicate that inoculated plants (M+) have on average greater (or lesser) enzymatic activity or soil nutritional content than not inoculated plants (M−).

**Table 1 jof-09-00421-t001:** Results of two-way repeated measures ANOVA for the survival for the four tree species. Main factors: Time, AMF treatments, and their interaction. Degrees of freedom (d.f.), F-values, and *p*-values are shown.

Species	Factor	d.f.	F-Value	*p*-Value
** *Cryptocarya alba* **	Time	4, 40	148.53	<0.001
	Treatment	1, 40	144.11	<0.001
	Time × Treatment	4, 40	9.1	<0.001
** *Kageneckia oblonga* **	Time	4, 40	123.72	<0.001
	Treatment	1, 40	30.41	<0.001
	Time × Treatment	4, 40	3.17	0.72
** *Peumus boldus* **	Time	4, 40	110.96	<0.001
	Treatment	1, 40	7.04	<0.05
	Time × Treatment	4, 40	4	<0.05
** *Quillaja saponaria* **	Time	4, 40	124.68	<0.001
	Treatment	1, 40	20.48	<0.001
	Time × Treatment	4, 40	1.88	0.133

**Table 2 jof-09-00421-t002:** Results of two-way repeated measures ANOVA for photosynthesis for the four tree species. Main factors: Time, AMF treatments, and their interaction. Degrees of freedom (d.f.), F-values, and *p*-values are shown.

Species	Factor	d.f.	F-Value	*p*-Value
** *Cryptocarya alba* **	Time	4, 40	26.30	<0.001
	Treatment	1, 40	286.62	<0.001
	Time × Treatment	4, 40	28.36	<0.001
** *Kageneckia oblonga* **	Time	4, 40	28.37	<0.001
	Treatment	1, 40	103.88	<0.001
	Time × Treatment	4, 40	3.39	0.054
** *Peumus boldus* **	Time	4, 40	13.42	<0.001
	Treatment	1, 40	22.62	<0.001
	Time × Treatment	4, 40	7.02	<0.001
** *Quillaja saponaria* **	Time	4, 40	67.84	<0.001
	Treatment	1, 40	269.57	<0.001
	Time × Treatment	4, 40	19.54	<0.001

**Table 3 jof-09-00421-t003:** Results of two-way repeated measures ANOVA on enzymes for the four tree species. Main factors: Time, AMF treatments, and their interaction. Degrees of freedom (d.f.), F-values, and *p*-values are shown.

** *(a) Cryptocarya alba* **				
**Enzyme**	**Factor**	**d.f**	**F-value**	***p*-value**
Dehydrogenase	Time	2, 12	212.64	<0.001
	Treatment	1, 12	591.50	<0.001
	Time × Treatment	2, 12	240.93	<0.001
β-Glucosidase	Time	2, 12	730.94	<0.001
	Treatment	1, 12	1410.62	<0.001
	Time × Treatment	2, 12	649.06	<0.001
Urease	Time	2, 12	87.30	<0.001
	Treatment	1, 12	90.00	<0.001
	Time × Treatment	2, 12	29.10	<0.001
** *(b) Kageneckia oblonga* **				
**Enzyme**	**Factor**	**d.f.**	**F-value**	***p*-value**
Dehydrogenase	Time	2, 12	126.25	<0.001
	Treatment	1, 12	126.87	<0.001
	Time × Treatment	2, 12	63.42	<0.001
β-Glucosidase	Time	2, 12	871.79	<0.001
	Treatment	1, 12	368.58	<0.001
	Time × Treatment	2, 12	220.28	<0.001
Urease	Time	2, 12	45.89	<0.001
	Treatment	1, 12	43.61	<0.001
	Time × Treatment	2, 12	12.52	<0.01
** *(c) Peumus boldus* **				
**Enzyme**	**Factor**	**d.f.**	**F-value**	***p*-value**
Dehydrogenase	Time	2, 12	204.97	<0.001
	Treatment	1, 12	72.00	<0.001
	Time × Treatment	2, 12	42.84	<0.001
β-Glucosidase	Time	2, 12	401.77	<0.001
	Treatment	1, 12	47.69	<0.001
	Time × Treatment	2, 12	51.80	<0.001
Urease	Time	2, 12	692.69	<0.001
	Treatment	1, 12	1508.10	<0.001
	Time × Treatment	2, 12	900.46	<0.001
** *(d) Quillaja saponaria* **				
**Enzyme**	**Factor**	**d.f.**	**F-value**	***p*-value**
Dehydrogenase	Time	2, 12	111.65	<0.001
	Treatment	1, 12	2.53	0.41
	Time × Treatment	2, 12	0.30	0.98
β-Glucosidase	Time	2, 12	3678.07	<0.001
	Treatment	1, 12	399.87	<0.001
	Time × Treatment	2, 12	286.62	<0.001
Urease	Time	2, 12	94.04	<0.001
	Treatment	1, 12	0.01	0.99
	Time × Treatment	2, 12	0.02	0.98

**Table 4 jof-09-00421-t004:** Results of *t*-test of macro-nutrients for the four tree species inoculated (M+) and not inoculated (M−) with AMF. Degrees of freedom (d.f.), *t*-values, and *p*-values are shown. Significant differences are denoted in red color.

	d.f.	*t*-Test	*p*-Value
** *(a) Cryptocarya alba* **			
Nitrogen	8	18.62	<0.0001
Phosphorus	8	11.41	<0.0001
Potassium	8	11.56	<0.0001
** *(b) Kageneckia oblonga* **			
Nitrogen	8	10.96	<0.0001
Phosphorus	8	14.62	<0.0001
Potassium	8	4.10	=0.0034
** *(c) Peumus boldus* **			
Nitrogen	8	8.97	<0.0001
Phosphorus	8	3.92	=0.0044
Potassium	8	5.10	<0.0001
** *(d) Quillaja saponaria* **			
Nitrogen	8	1.25	=0.2460
Phosphorus	8	0.25	=0.8031
Potassium	8	0.39	=0.7010

## Data Availability

The data presented in this study related with survival, photosynthesis, enzymatic activity and soil nutrients are available on request from the corresponding author.

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
