# Peer review of "Assessing the Importance of Native Mycorrhizal Fungi to Improve Tree Establishment after Wildfires"

_jof, 2023, doi:10.3390/jof9040421_

Round 1
Reviewer 1 Report
Authors Atala et al. wrote a good paper on important topic. The text is easy to follow and is well organized. However, I have some questions/suggestions:
Abstract:
- line 14: I would not put "Mycorrhizal fungi are key microorganisms, as they can ..." but something like "Myc. fungi may be key microorganisms to help plants ..."
Introduction:
- line 81: "... search for possible mechanisms". For me this is unclear. As much as I understood, this paper does not deal with mechanisms that are going on during your experiment. You only added AMF inoculum and registered the consequences.
- line 82: Instead of "Information on these mycorrhizal fungi could ...": I think it would be more correct to say something like "Information on the efficacy of AMF application to the seedlings of studied plant species..." or something like that.
Materials and methods:
- line 101: Did you check what was AMF root colonization percentage of the sampled roots? Or spore number in the soil? If yes, this should be added.
- line 121: Was 5% the limit and all the higher than that was considered as M+?
- line 125: Native soil - did you check the presence of native AMF in the soil where you planted your plants?
Results:
- Figure 1: The number of survived individuals decreased continuously. Although you found out that plants in M+ treatment showed significantly higher survival, one could presume that after 24 months plant continued to dye. Also, was the final number of survived plants after 24 months in M+ treatment enought to say that it makes sense to apply AMF? Did you at the end of experiment estimate the status of the survived plants - did they only survive or they were in good status?
- Figure 3. Does the left side of the graph refer to M- and the right side to the M+? If yes, please sign it.
- Did you at any stage estimate root colonization of M+ and M- plants? It is quite probable that M- plants were colonized after some time by the AMF present on site. Were M+ and M- plants significantly different in root colonization after transplantation to the forest? If you have these data, please show them and discuss them.
Discussion:
- line 246: You say that negative effect of myc. fungi has been reported in highly colonized plants, leading to parasitism. But I think that you should show that P. boldus root colonization was very high in order to refer to this. Do you have this information? If not, I suggest to omit this sentence because it is too speculative.
- line 272: "... positive nutritional effect..."; Nutritional effect in this study is found only indirectly and potentially. I would not use this phrase in this paper.
- line 281: I would omit this sentence. Same AMF species are not evidence of CMN. And it is not crucial for this paper.
Author Response
REVIEWER 1
Commentary: Authors Atala et al. wrote a good paper on important topic. The text is easy to follow and is well organized. However, I have some questions/suggestions.
Reply: Many thanks for this general positive appreciation about our study.
Commentary: line 14: I would not put "Mycorrhizal fungi are key microorganisms, as they can ..." but something like "Myc. fungi may be key microorganisms to help plants ..."
Reply: Thanks for this comment, we modified the draft accordingly.
Commentary: line 81: "... search for possible mechanisms". For me this is unclear. As much as I understood, this paper does not deal with mechanisms that are going on during your experiment. You only added AMF inoculum and registered the consequences.
Reply: We agree with this commentary, and the sentence was modified.
Commentary: line 82: Instead of "Information on these mycorrhizal fungi could ...": I think it would be more correct to say something like "Information on the efficacy of AMF application to the seedlings of studied plant species..." or something like that.
Reply: Thanks for this comment, we modified the draft accordingly.
Commentary: line 101: Did you check what was AMF root colonization percentage of the sampled roots? Or spore number in the soil? If yes, this should be added.
Reply: We are very grateful for this commentary since we have now the opportunity to clarify this point. Indeed, after seeing your commentary we realized that this information is pivotal to understand the methodology but was omitted in the previous version. In this revised version of the manuscript, we added a supplementary table where is indicated the percentage of mycorrhizal structures after application of inoculation (M+) as well as in the control treatment (M-), in individuals that posteriorly were allocated to the transplant experiment.
Commentary: line 121: Was 5% the limit and all the higher than that was considered as M+?
Reply: This sentence was rewrite in order to improve the clarity.
Commentary: line 125: Native soil - did you check the presence of native AMF in the soil where you planted your plants?
Reply: Regrettably we don't analyze the soil previously the assay, but we want to remember that the site where the assay was conducted was burned by a wildfire, and possibly the most of mycorrhizal structures were destroyed. On the other hand, our results showed that independently the "potential" presence of mycorrhizal structures in the soil of study site, the direct root inoculation in seedlings had a most important effect than a potential posterior natural inoculation.
Commentary: Figure 1: The number of survived individuals decreased continuously. Although you found out that plants in M+ treatment showed significantly higher survival, one could presume that after 24 months plant continued to dye. Also, was the final number of survived plants after 24 months in M+ treatment enought to say that it makes sense to apply AMF? Did you at the end of experiment estimate the status of the survived plants - did they only survive or they were in good status?
Reply: We respectfully disagree with this commentary, since as is possible to see in the figure 1, in two of four tree species the survival was stable in the last 12 months, and in another two tree species the survival decreases smoothly. Thus, we don’t think the survival showed a "continuous" decreasing with time. On the other hand, every individual was visually assessed in every measure to sure that a givel individuals was correctly allocated to alive or dead.
Commentary: Figure 3. Does the left side of the graph refer to M- and the right side to the M+? If yes, please sign it.
Reply: Not necessarily. In fact, the size effect graphs are arbitrary in the assignation of "positive or negative" effect according to the way how were calculated. Nevertheless, in the revised version we added more details in the caption of the figure.
Commentary: Did you at any stage estimate root colonization of M+ and M- plants? It is quite probable that M- plants were colonized after some time by the AMF present on site. Were M+ and M- plants significantly different in root colonization after transplantation to the forest? If you have these data, please show them and discuss them.
Reply: As we indicated above, we validated the inoculation method and the colonization status before transplant experiment. We don’t record the colonization after the transplant, and probably that individuals allocated to "control" treatment were naturally inoculated with time. Nevertheless, the most relevant for restoration purposes is the early inoculation as well as was showed by results.
However, could be very interesting to know how fast the natural infection in field is or how is the kinetic of infection. We will have in mind this question for a future study.
Commentary: line 246: You say that negative effect of myc. fungi has been reported in highly colonized plants, leading to parasitism. But I think that you should show that P. boldus root colonization was very high in order to refer to this. Do you have this information? If not, I suggest to omit this sentence because it is too speculative.
Reply: In the revised version of the manuscript, we rewrite this paragraph, and this sentence is supported by the new information from the supplementary table 1.
Commentary: line 272: "... positive nutritional effect..."; Nutritional effect in this study is found only indirectly and potentially. I would not use this phrase in this paper.
Reply: In the revised version of the manuscript, we added a new dataset related with soil macronutrients (N, P, and K). Thus, we performed a new Figure, statistical analysis, and table, which allow to support the aspects related with the nutritional status improved by inoculation.
Reviewer 2 Report
Assessing the importance of soil mycorrhizal fungi to improve tree establishment after wildfires
Cristian Atala1, Sebastián A. Reyes1 and Marco A. Molina-Montenegro2,3*
This an excellent paper in an important area of restoration research that provides solid, original data with good insights based upon it. It provides a sound basis upon which to build with additional research in the Chilean MTE. I recommend accepting this paper with minor revisions.
Overall, this is a well-written paper. It would be beneficial to discuss post-fire approaches to assuring the presence of mycorrhizae employed in other MTEs, particularly in California, for example. The use of mycorrhizae colonized soil, for example, could be mentioned as an alternate approach (for example see Gerrits et al., 2023). Ideas regarding how to conduct largescale inoculation would be helpful, and the use of local or site-specific inoculum should be considered. The authors should archive samples of the mycorrhizae from each tree species for future species-level identification.
There are minor wording changes that I recommend, and I am attaching a marked-up copy of the manuscript with suggested changes indicated. I urge the authors to review the narrative thoroughly as there are some sentences that are difficult to understand and there are probably other minor changes that need to be addressed and modified. A number of scientific names are not italicized in the references citations, and the authors should make certain that the listing of papers is consistent in format for all of the references. Citation of each reference in the narrative should be verified.
Gerrits, G.M. et al. 2023. Synthesis on the effectiveness of soil translocation for plant community restoration. Journal of Applied Ecology DOI: 10.1111/1365-2664.14364
https://doi.org/10.1111/1365-2664.14364

Author Response
Commentary: This an excellent paper in an important area of restoration research that provides solid, original data with good insights based upon it. It provides a sound basis upon which to build with additional research in the Chilean MTE. I recommend accepting this paper with minor revisions.
Reply: Many thanks for your commentaries.
Commentary: Overall, this is a well-written paper. It would be beneficial to discuss post-fire approaches to assuring the presence of mycorrhizae employed in other MTEs, particularly in California, for example. The use of mycorrhizae colonized soil, for example, could be mentioned as an alternate approach (for example see Gerrits et al., 2023). Ideas regarding how to conduct largescale inoculation would be helpful, and the use of local or site-specific inoculum should be considered. The authors should archive samples of the mycorrhizae from each tree species for future species-level identification.
Reply: Many thanks for this commentary. In the revised version we have added a new paragraph related with the study conducted by Gerrits et al. 2023. In fact, we indicate that our approach could be a complementary strategy to other, for example, that commented in the article by Gerrits et al. 2023.
Commentary: There are minor wording changes that I recommend, and I am attaching a marked-up copy of the manuscript with suggested changes indicated. I urge the authors to review the narrative thoroughly as there are some sentences that are difficult to understand and there are probably other minor changes that need to be addressed and modified. A number of scientific names are not italicized in the reference’s citations, and the authors should make certain that the listing of papers is consistent in format for all of the references. Citation of each reference in the narrative should be verified.
Reply: In the new version of the manuscript, we have conducted a full revision in order to improve the flow and clarity in the main ideas. In addition, we done a careful check in the references to avoid the typo mistakes.
Commentary: Gerrits, G.M. et al. 2023. Synthesis on the effectiveness of soil translocation for plant community restoration. Journal of Applied Ecology DOI: 10.1111/1365-2664.14364
https://doi.org/10.1111/1365-2664.14364
Reply: Many thanks for recommend us this excellent paper.
Reviewer 3 Report
Dear authors,
I am writing to express my concern regarding the manuscript.
First of all, the title should be changed. Mycorrhizal fungi are present in soil only. It is unnecessary to call them soil mycorrhizal fungi.
A more specific title could be 'Assessing the importance of native arbuscular mycorrhizal fungi (AMF) to improve tree establishment after wildfires in the Chilean matorral area?
My main concern is how the authors are certain that the seedlings are infected with AMF. There is no data indicating the percentage of infection rate, and even the method used to obtain the inoculum containing mycorrhizal structures, as indicated on page 1 lines 100-101, raises questions about how the authors ensured that they obtained AMF propagules to use in the study. It would have been beneficial if the authors had examined the AMF infection rate in their seedlings to prove that there was a high infection rate in the mother trees, and vice versa.
Authors said in page one line 120-123 but how many of .....?
and after 5 months old seedlings were transplanted into the field how to make sure that this is effected from amf inoculum or from filed AMF exist in the soil/plots?
line 265 ; sue to economic? due to ?
line 249; explain for general readers why these 3 enzymes are key enzymes?
Author Response
Commentary: First of all, the title should be changed. Mycorrhizal fungi are present in soil only. It is unnecessary to call them soil mycorrhizal fungi. A more specific title could be 'Assessing the importance of native arbuscular mycorrhizal fungi (AMF) to improve tree establishment after wildfires in the Chilean matorral area?
Reply: Thanks for this commentary, and in the revised version we have modified the title accordingly.
Commentary: My main concern is how the authors are certain that the seedlings are infected with AMF. There is no data indicating the percentage of infection rate, and even the method used to obtain the inoculum containing mycorrhizal structures, as indicated on page 1 lines 100-101, raises questions about how the authors ensured that they obtained AMF propagules to use in the study. It would have been beneficial if the authors had examined the AMF infection rate in their seedlings to prove that there was a high infection rate in the mother trees, and vice versa.
Reply: We are very grateful for this commentary since we have now the opportunity to clarify this point. Indeed, after seeing your commentary we realized that this information is pivotal to understand the methodology but was omitted in the previous version. In this revised version of the manuscript, we added a supplementary table where is indicated the percentage of mycorrhizal structures after application of inoculation (M+) as well as in the control treatment (M-), in individuals that posteriorly were allocated to the transplant experiment.
Commentary: After 5 months old seedlings were transplanted into the field how to make sure that this is effected from amf inoculum or from filed AMF exist in the soil/plots?
Reply: My apologies but I understood this question. Nevertheless, the evaluation or verification to assess if the treatment M+ or M- in the individuals used for the field experiments was few days before to be transplanted into the field.
Commentary: line 265; sue to economic? due to ?
Reply: Thanks. This mistake was corrected.
Commentary: line 249; explain for general readers why these 3 enzymes are key enzymes?
Reply: In the revised version we have added a new paragraph highlighting the importance of these three enzymes. See L 164-169.
Reviewer 4 Report
REVIEW REPORT
The study assesses the effect of mycorrhizal inoculation on survival and photosynthesis along two years after a fire event in four native dominant woody plant species, namely, Peumus boldus, Quillaja saponaria, Cryptocarya alba, and Kageneckia oblonga in the Mediterranean ecosystem. The results suggest that mycorrhizal fungi could increase fitness of plants used in restoration initiatives after severe disturbances such as fires and, consequently, should be considered for restoration programs of native species in threatened Mediterranean ecosystem.
MAJOR POINTS OF CONCERN
· What AMF species/strains were used to inoculate the target tree species? An assessment of AMF species could have added enormous value to the study. Taxonomic identification (morphological + DNA based) could have given an idea what all species of AMF work best in the restoration of native tree species in threatened ecosystem.
· L-101-102: I have doubts about the effectiveness of the method of inoculation of AMF that the authors have used (also because the colonisation percentage and pictures confirming colonisation and infection propagules have not been submitted). Relevant studies have also not been cited.
· L 102-103: 75g of roots in 100 ml of water amount to what proportion of fungal propagules (spores, etc.) in the solution?? Were the number of propagules estimated prior inoculation. Amount of solution received by each seedling is also not clear. How was homogeneity ensured?
· Colonisation pictures are not supplied in the manuscript.
· L 120-121: Authors have just investigated the effectiveness of the AMF inoculation. But the extent of colonisation, i.e., Colonisation percentage, is not calculated anywhere. How the tree species differed in their symbiotic potential is a major factor in a study of this nature.
· Estimation of three enzymes does not provide a window into the Nutritional effect of the mycorrhizal inoculation on the soil. To corroborate the enzyme activities with the nutritional effects of AMF, assessment of nutrient (micro and macro) profile should have been done.
· In the M & M section, L-107 section “Effects of mycorrhiza on survival and photosynthesis of trees” does not cover the effects per se. Instead, the section should be sub-headed as ‘Assessment of Gas-exchange parameters’.
· Relevance of the study is limited to a very local domain of audience, concerning MTEs of Chile, however, the concern raised, and a prospective solution offered in the form of mycorrhizal symbiosis, that could be used as a tool for increasing restoration success after fires, is appreciated. But, I feel, the parameters investigated in the study (Gas exchange and soil enzyme activities) to gauge the survival and performance of native woody plants after fire are not enough to claim that administration with AMF was found improved growth, establishment, and nutrient availability in the soil.
Author Response
Commentary: What AMF species/strains were used to inoculate the target tree species? An assessment of AMF species could have added enormous value to the study. Taxonomic identification (morphological + DNA based) could have given an idea what all species of AMF work best in the restoration of native tree species in threatened ecosystem.
Reply: We agree with the interesting about to know the identity of strains by molecular or taxonomic methods. Nevertheless, was not the aim of our study. In fact, to accomplish aim of know if the native mycorrhizal structures exert a positive role on the establishment of the host-tree species, is irrelevant to know the identity of the strain. Thus, we are looking for the "effects", at least at this state of the research, but not rule out that the next steps could be related with the selection of specific strains and hence we should conduct molecular or taxonomic identification.
Commentary: L-101-102: I have doubts about the effectiveness of the method of inoculation of AMF that the authors have used (also because the colonisation percentage and pictures confirming colonisation and infection propagules have not been submitted). Relevant studies have also not been cited.
Reply: The methodology used to transfer mycorrhizal structures have been previously used in other studies. In fact, in the last time several authors have used "rhizospheric soil" to transfer not only mycorrhizas, but nutrients, bacteria, other fungi and minerals.
On the other hand, we are very grateful for this commentary since we have now the opportunity to clarify this point. Indeed, after seeing your commentary we realized that this information is pivotal to understand the methodology but was omitted in the previous version. In this revised version of the manuscript, we added a supplementary table where is indicated the percentage of mycorrhizal structures after application of inoculation (M+) as well as in the control treatment (M-), in individuals that posteriorly were allocated to the transplant experiment.
Commentary: L 102-103: 75g of roots in 100 ml of water amount to what proportion of fungal propagules (spores, etc.) in the solution?? Were the number of propagules estimated prior inoculation. Amount of solution received by each seedling is also not clear. How was homogeneity ensured?
Reply: In the revised version we have clarified that individuals (M+) received 100mL per day and those allocated to M- treatment received the same amount but without inoculum (only tap-water). Although, we have not estimated the number of propagules inoculated, the new information (Suppl. Table 1), suggest that the colonisation assessed as frequency of occurrence was successful and similar among all tree species.
Commentary: L 120-121: Authors have just investigated the effectiveness of the AMF inoculation. But the extent of colonisation, i.e., Colonisation percentage, is not calculated anywhere. How the tree species differed in their symbiotic potential is a major factor in a study of this nature.
Reply: As commented above, in the revised version we have included new information (see Suppl. Table 1) about the frequency of occurrence of mycorrhizal structure in each tree species for both treatments (M+ and M-).
Commentary: Estimation of three enzymes does not provide a window into the Nutritional effect of the mycorrhizal inoculation on the soil. To corroborate the enzyme activities with the nutritional effects of AMF, assessment of nutrient (micro and macro) profile should have been done.
Reply: Thanks for this commentary. As was requested by this reviewer, in the revised version we have added a new dataset related with the content of macronutrients in the soil. Thus, in the revised version of the manuscript we have added a new table (Table 4) and a new figure (Figure 4).
Commentary: In the M & M section, L-107 section “Effects of mycorrhiza on survival and photosynthesis of trees” does not cover the effects per se. Instead, the section should be sub-headed as ‘Assessment of Gas-exchange parameters’.
Reply: This sub-head was modified accordingly to the reviewer´s comments.
Commentary: Relevance of the study is limited to a very local domain of audience, concerning MTEs of Chile, however, the concern raised, and a prospective solution offered in the form of mycorrhizal symbiosis, that could be used as a tool for increasing restoration success after fires, is appreciated. But, I feel, the parameters investigated in the study (Gas exchange and soil enzyme activities) to gauge the survival and performance of native woody plants after fire are not enough to claim that administration with AMF was found improved growth, establishment, and nutrient availability in the soil.
Reply: In the discussion section we toned-down the suggestion about "effect of mycorrhizal" to solve the problem of restoration in areas affected by wildfires. In this revised version, we proposed that mycorrhizal structures could be a methodology complementary to others to help to improve the success of restoration.
Reviewer 5 Report
Dear Authors,
Your manuscript is very interesting and has the potential to be published. However, I would like to give some comments and suggestions for you to improve your manuscript.
Introduction Line 53 and several others in the manuscript - please fix the agreement mistake MTE, it is supposed to be written as MTEs
Materials and Methods Line 95 - it is supposed to be four genera and not six genera.
Materials and Methods for section Nutritional effect of the mycorrhizal inoculation on the soil, I have detected a high similarity between your writings with methods described by Barrera et al (2022) article entitled Biological soil crusts as ecosystem engineers in Antarctic ecosystem. Please revise the sentences accordingly.
Materials and Methods line 150 - wrong spelling of Nitrophenil-B-D-glucopyranoside
Results line 175 - I would suggest writing species names in full at the beginning of a new section.
Author Response
Commentary: Your manuscript is very interesting and has the potential to be published. However, I would like to give some comments and suggestions for you to improve your manuscript.
Reply: Many thanks for the positive general appreciation about our study.
Commentary: Introduction Line 53 and several others in the manuscript - please fix the agreement mistake MTE, it is supposed to be written as MTEs.
Reply: Done.
Commentary: Materials and Methods Line 95 - it is supposed to be four genera and not six genera.
Reply: Thanks. Corrected.
Commentary: Materials and Methods for section Nutritional effect of the mycorrhizal inoculation on the soil, I have detected a high similarity between your writings with methods described by Barrera et al (2022) article entitled Biological soil crusts as ecosystem engineers in Antarctic ecosystem. Please revise the sentences accordingly.
Reply: In the revised version of the manuscript, we have rewritten all this section.
Commentary: Materials and Methods line 150 - wrong spelling of Nitrophenil-B-D-glucopyranoside .
Reply: Thanks. Corrected.
Commentary: Results line 175 - I would suggest writing species names in full at the beginning of a new section.
Reply: Done.
Round 2
Reviewer 3 Report
table description for every table should be prior table?
keywords; repeated between Arbuscular mycorrhiza and Arbuscular mycorrhiza fungi. delete one and replace with mycorrhization?
line 152; were? were done?
Results.
line 197...more marked? replace with remarkably?
line 244; This was so for...? This was the same with..?
line 269; tab paragraph
line 279; In contrast..?
line 301; change to.. Those trees that exhibited improved nutritional status levels were the same ones that demonstrated an increase in enzymatic activity. ???
Table S1. +/- mean Standard deviation? and for M-, SD is apparently higher than mean? what would happened with this figures?
Author Response
Dear Reviewer,
Many thanks for your time spent to our manuscript. In the revised version we have attended all your inquiries.
Commentary 1: table description for every table should be prior table?
Reply: Thanks for notice this detail. In the revised version it was corrected.
Commentary 2: keywords; repeated between Arbuscular mycorrhiza and Arbuscular mycorrhiza fungi. delete one and replace with mycorrhization?
Reply: Thanks. In the revised version was deleted the keyword: Arbuscular mycorrhiza.
Commentary 3: line 152; were? were done?
Reply: Corrected.
Commentary 4: line 197...more marked? replace with remarkably?
Reply: Replaced.
Commentary 5: line 244; This was so for...? This was the same with?
Reply: Corrected.
Commentary 6: line 279; In contrast?
Reply: Done.
Commentary 7: line 301; change to.. Those trees that exhibited improved nutritional status levels were the same ones that demonstrated an increase in enzymatic activity???
Reply: Thanks. The sentence was modified accordingly to this commentary.
Commentary 8: Table S1. +/- mean Standard deviation? and for M-, SD is apparently higher than mean? what would happened with this figures?
Reply: Yes, indeed. This happen since the number of samples is low and the response variable is in percentage. For example, if we have five samples with one of them recording 60% but the others with 0%, the mean will be low but the S.D., very high.
Reviewer 4 Report
Answers to the raised questions are not satisfactory. The AM %colonization data that has been presented is devoid of statistical analysis and has all the more made the presentation of the study more confusing. What are these fungal endophytes? And how come AM structures are present in the trees which were not subjected to any AMF treatment? (-M) Moreover, pictures of colonization are not provided despite pointing it out in one of the comments raised.
Furthermore, nutrient estimation (N,P,K) has been incorporated in the study but its relevance with respect to the obtained results has not been discussed anywhere in the discussion section.
Author Response
Commentary: Answers to the raised questions are not satisfactory. The AM %colonization data that has been presented is devoid of statistical analysis and has all the more made the presentation of the study more confusing. What are these fungal endophytes? And how come AM structures are present in the trees which were not subjected to any AMF treatment? (-M) Moreover, pictures of colonization are not provided despite pointing it out in one of the comments raised.
Reply: In the revised version of the manuscript, we have added a statistical analysis (Randomization test) to compare the frequency of occurrence of mycorrhizal structures and fungal endophytes as requested by the reviewer. Thus, the new supplementary table was modified accordingly to this commentary.
On the other hand, effectively in some samples of roots (n = 1 or 2 per tree species) was recorded the presence of some mycorrhizal structure, but this was less than 5% and always significantly lower than those trees in M+ treatment. Lastly, as we pointed in the former rebuttal letter, our aim was not characterizing (by morphologic or molecular approaches) the different mycorrhizal found in our study, since we approach is functional considering the whole community. Our apologies, but we have no interest to conduct additional experiments related with the characterization trough pictures.
Commentary: Furthermore, nutrient estimation (N,P,K) has been incorporated in the study but its relevance with respect to the obtained results has not been discussed anywhere in the discussion section.
Reply: We are partially agree with this commentary, because this new data set was not only requested by other reviewers, but makes sense together with the enzymatic activity and both are linked with higher photosynthesis and survival, as indicated in the discussion section.